# Baseline patterns of infection in regions of Benin, Malawi and India seeking to interrupt transmission of soil transmitted helminths (STH) in the DeWorm3 trial

**The DeWorm3 Trials Team**¶*

¶Membership of the DeWorm3 Trials Team is provided in the S2 Text.
* walson@uw.edu

**Data Availability Statement:** Data cannot be shared publicly because the study remains blinded

## Abstract

Global efforts to control morbidity associated with soil-transmitted helminth infections (STH) have focused largely on the targeted treatment of high-risk groups, including children and pregnant women. However, it is not clear when such programs can be discontinued and there are concerns about the sustainability of current STH control programs. The DeWorm3 project is a large multi-country community cluster randomized trial in Benin, India and Malawi designed to determine the feasibility of interrupting the transmission of STH using community-wide delivery of mass drug administration (MDA) with anthelmintics over multiple rounds. Here, we present baseline data and estimate key epidemiological parameters important in determining the likelihood of transmission interruption in the DeWorm3 trial. A baseline census was conducted in October-December 2017 in India, November-December 2017 in Malawi and in January-February 2018 in Benin. The baseline census enumerated all members of each household and collected demographic data and information on occupation, assets, and access to water, sanitation and hygiene (WASH). Each study site was divided into 40 clusters of at least 1,650 individuals per cluster. Clusters were randomized to receive twice yearly community-wide MDA with albendazole (GSK) targeting eligible individuals of all ages (20 clusters), or to receive the standard-of-care deworming program targeting children provided in each country. In each site, a randomly selected group of 150 individuals per cluster (6,000 total per site) was selected from the baseline census using stratified random sampling, and each individual provided a single stool sample for analysis of STH infection using the Kato-Katz technique. Study site, household and individual characteristics were summarized as appropriate. We estimated key epidemiological parameters including the force of infection and the degree of parasite aggregation within the population. The DeWorm3 sites range in population from 94,969 to 140,932. The population age distribution varied significantly by site, with the highest proportion of infants and young children in Malawi and the highest proportion of adults in India. The baseline age- and cluster-weighted prevalence, as measured by Kato-Katz, varied across sites and by species, Baseline hookworm prevalence in India was 21.4% (95% CI: 20.4–22.4%), while prevalence of *Ascaris*

to outcome data. Data are available from the DeWorm3 Institutional Data Access Committee (contact via Barbra Richardson, barbrar@uw.edu) for researchers who meet the criteria for access to these data.

**Funding:** The DeWorm3 Project is funded by a grant from the Bill & Melinda Gates Foundation (OPP1129535) TL, JLW. The funders had no role in study design, data collection and analysis, decision to publish, or preparation of the manuscript.

**Competing interests:** The authors have declared that no competing interests exist.

and *Trichuris* by Kato-Katz was low (0.1% and 0.3% overall). In Malawi, the overall age- and cluster-weighted STH prevalence was 7.7% (95% CI: 7.1–8.4%) predominantly driven by hookworm infections (7.4%) while *Ascaris* (0.1%) and *Trichuris* (0.3%) infections were rare. In Benin, the overall age- and cluster-weighted prevalence was significantly lower (5.6%, 95% CI: 5.1–6.2%) and *Ascaris* (2.0%, 95% CI: 1.6–2.3%) was more common than in other sites. *Ascaris* infections were more likely to be moderate- or heavy-intensity (43.7%, unweighted) compared to hookworm (5.0%). The force of infection for hookworm was highest in adults in India and Malawi but appeared relatively stable across age groups in Benin. These data demonstrate the significant variability between the sites in terms of demography, socio-economic status and environmental characteristics. In addition, the baseline prevalence and intensity data from DeWorm3 suggest that each site has unique epidemiologic characteristics that will be critical in determining correlates of achieving STH transmission interruption in the DeWorm3 trial.

**Trial registration:** The trial was registered at ClinicalTrials.gov (NCT03014167).

## Author summary

Intestinal parasitic worms, soil-transmitted helminths, are among the most common infectious organisms of humans. In many low-resource settings, these infections result in considerable morbidity, including reductions in childhood growth and development, increased risk of anemia and reductions in future educational achievement and income earning potential. The current global strategy for controlling these infections is through routine deworming of school and pre-school aged children as well as pregnant women. Since many adults and non-school going children are infected with these parasites, the current strategy does not prevent the rapid reinfection of individuals despite repeated treatment. The DeWorm3 trial is a large multi-country trial being conducted in Benin, India and Malawi to test the feasibility of using mass drug administration of deworming medications to all individuals in a community to interrupt these infections in some geographic areas. Here we present baseline data from these communities and estimate the transmission potential of these infections at each of the sites.

## Introduction

Soil-transmitted helminth infections (STH) remain among the most common infections of human populations globally, resulting in substantial morbidity and loss of productivity in low-resource settings[1]. Current World Health Organization (WHO) guidelines focus on the control of STH associated morbidity using mass drug administration (MDA) of anthelmintics delivered to groups at high risk of morbidity, including children and women of reproductive age[2]. Many national governments have adopted this strategy, and in 2017, more than 743 million people received preventive chemotherapy for soil-transmitted helminthiases (188 million preschool-aged children (PSAC), 410.1 million school-aged children (SAC) and 127.9 million women of reproductive age)[3]. In countries where high coverage has been achieved, substantial reductions in high and moderate-intensity infections, with associated decreases in STH associated morbidity, have been documented[4].

Despite reductions in morbidity, in areas of endemic infection the current approach focusing on morbidity control may not reduce the overall prevalence of infection below levels

needed to interrupt transmission[5, 6]. Poor sanitation and hygiene, coupled with large reservoirs of infection in untreated individuals (including non-pregnant adults and children missed by current programs), serve to perpetuate transmission in many low-resource settings. Settings where hookworm is the dominant infection may be particularly challenging, since hookworm burden is typically highest among adults. As a result, deworming programs are faced with the prospect of having to continue to treat high-risk populations repeatedly, until such time as economic and infrastructural development (including improvements in water, sanitation and hygiene; WASH) can effectively prevent continued transmission[7, 8].

In the absence of massive economic and infrastructural development, it may still be possible to interrupt the transmission of STH in some geographic settings. A recent trial in Kenya demonstrated that community-wide treatment was more effective than school-based treatment in reducing hookworm prevalence and intensity[9]. Mathematical models suggest that transmission interruption of STH may be possible with MDA alone, provided coverage of a substantial portion of the community is achieved repeatedly and individual compliance to treatment is high over many rounds of treatment[6]. The feasibility of such an approach depends on the underlying transmission intensity in a community, which is affected by a variety of factors including the demography of the targeted population, levels of access to clean water and sanitation, hygiene behaviours, population movement, and characteristics of the built and natural environment, as well as the intensity and frequency of prior deworming in the population[5].

The DeWorm3 project is a large multi-country community cluster randomized trial designed to determine the feasibility of interrupting the transmission of STH using community-wide delivery of MDA with anthelmintics over multiple rounds[10]. Here, we present baseline prevalence data and discuss the prevailing epidemiological patterns in the three study sites prior to the initiation of expanded community-wide MDA delivered through the DeWorm3 trial. Using baseline prevalence and intensity data from each study community, we describe the distribution of infections within each age group, estimate the number of heavy intensity infections as a proxy for morbidity, and calculate the estimated force of infection, or rate of new STH infections, within each geography.

## Methods

The protocol of the DeWorm3 trial has been previously published[10]. Briefly, DeWorm3 is a multi-site community cluster randomized trial being conducted in three study sites in Benin, Malawi and India (Fig 1). In each country, a predefined administrative unit containing at least 80,000 individuals was selected to participate in the trial. Selection was based on; 1) a history of having participated in at least five rounds of MDA for lymphatic filariasis (with albendazole and diethylcarbamazine (DEC) or ivermectin), 2) baseline estimates of STH endemicity and 3) support for participation from the respective national STH program manager and district or commune level officials. In both Malawi and Benin, the sites consist of geographically contiguous areas, with the Malawi site located in a rural area of Mangochi district, in the Southern region; and the Benin site comprising Comé town and the surrounding rural area in the Commune of Comé. In India, the study area consists of two geographically distinct sub-sites within the state of Tamil Nadu—a plains area in Timiri and a tribal region in Jawadhu Hills.

### Baseline census

In order to enumerate the entire population eligible for MDA within DeWorm3, a baseline census was conducted in October-December 2017 in India, November-December 2017 in Malawi and in January-February 2018 in Benin. Heads of household or other adult decision-

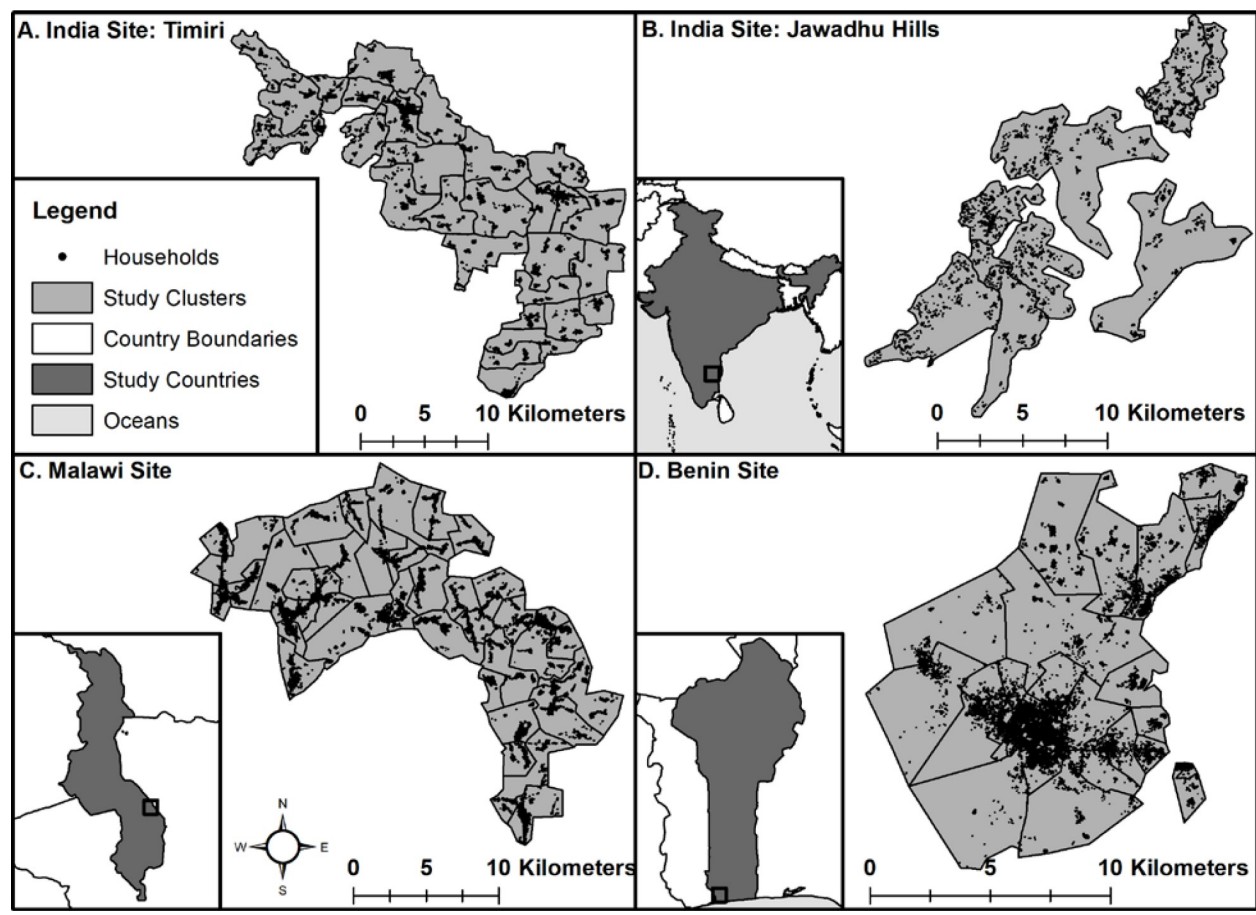

**Fig 1.** Maps of study sites in India (A-B), Malawi (C) and Benin (D) with cluster boundaries and GPS locations of all households in the Baseline Census overlaid (ArcGIS).

makers consented on behalf of their households to participate in the baseline census, which consisted of a combination of a questionnaire delivered to the household head plus observation of housing conditions by data collectors. Data were recorded electronically using Android phones with SurveyCTO software (Dobility, Inc; Cambridge, MA, USA and Ahmedabad, India). The baseline census enumerated all members of each household, including those considered migratory (having spent the majority of nights elsewhere in the past six months), and collected demographic data and information on occupation, assets, and access to WASH. The geospatial location of each dwelling was also collected and mapped using ArcGIS (ESRI, Redlands, CA, USA) and overlaid on satellite imagery using SASPlanet software (https://www.openhub.net/p/sasplanet) and QGIS (https://www.qgis.org). Each household was categorized based on surrounding population density. Population density per square kilometer was estimated by constructing a one km$^2$ buffer around each household in ArcGIS and summing to total number of individuals that fell within each household's buffer. For households near study area boundaries, areas of the buffer that fell outside of the study area were removed and the population density was calculated by the number of individuals falling within the buffer divided by the area of the buffer. Data on the construction material of the dwelling were recorded based on the data collector's observations. Households in the study area were visited up to three times during each site's census to maximize participation. Water sources and

sanitation facilities reported at each site were grouped and categorized according to the 2017 WHO/Unicef Joint Monitoring Programme methodology (none, unimproved, limited or basic)[11].

## Cluster demarcation

Each study site w[12]as divided into 40 clusters of at least 1,650 individuals per cluster based on the baseline census (Fig 2). Initial cluster boundaries were defined based on administrative units to which households reported belonging (sub-districts, blocks or villages). Small neighboring administrative units were combined to form clusters, and administrative units of population >6,500 were split into multiple clusters. Cluster boundaries that did not follow preexisting administrative boundaries were first based on geographical features such as rivers or roads where possible, and finally spatial location of households. At the final stage, households whose actual spatial location did not match their reported administrative units were reassigned to the cluster matching their location.

Clusters were randomized to receive twice yearly community-wide MDA with albendazole (GSK) targeting eligible individuals of all ages (20 clusters), or to receive the standard-of-care deworming program targeting children provided in each country: annual SAC- and PSAC-targeted MDA in Malawi, annual SAC-targeted MDA in Benin, and bi-annual SAC- and PSAC-targeted MDA in India (20 clusters).

## Baseline prevalence assessment

In each site, a randomly selected group of 150 individuals per cluster (6,000 total per site) was selected from the baseline census using stratified random sampling of PSAC (1 to 4 years), SAC (5 to 14 years) and adults (15 years and above) at a ratio of 1:1:3. For each cluster, a sampling list of 150 individuals was initially generated from the baseline census, and backup lists of 75 individuals were issued to replace participants who could not be located or refused to participate. Adult participants consented to be followed annually for the duration of the trial; parents provided consent on behalf of their children, and children provided assent based on relevant national guidelines. Participants were interviewed and completed a more detailed

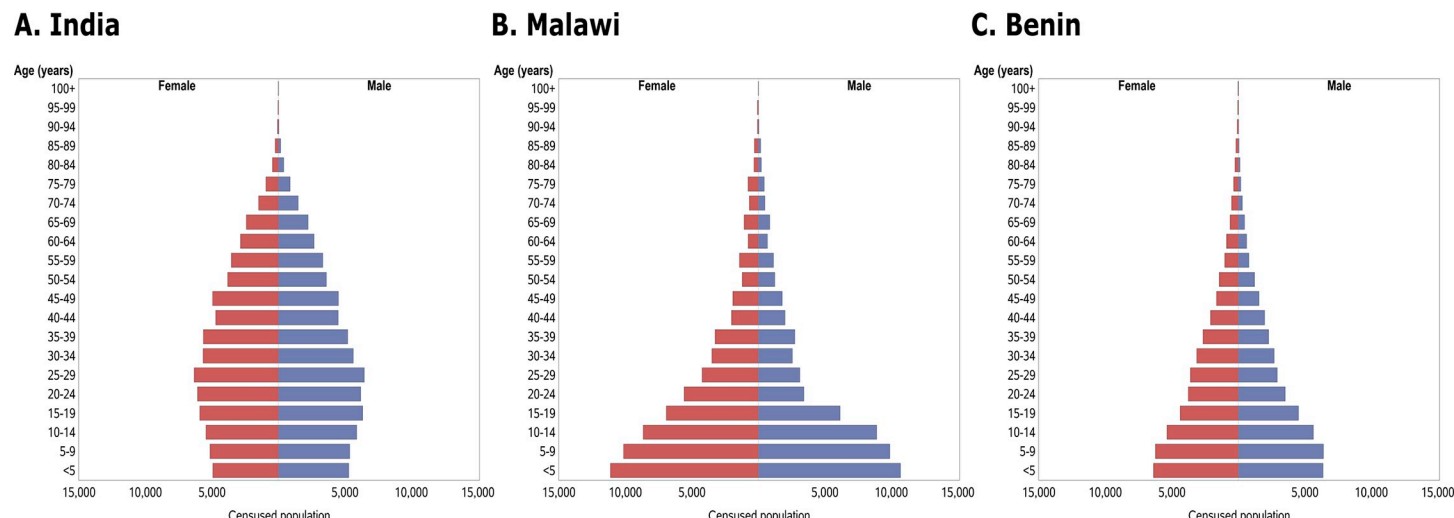

**Fig 2.** Population pyramids showing five-year bands of age (where available) and gender breakdown of the Baseline Census populations in India (A), Malawi (B) and Benin (C). These participants were broadly categorized as pre-school age children (PSAC, <5 years of age), school age children (SAC, 5–14 years) and adults (15+). In all figures, red represents the female population and blue represents males.

assessment of individual-level STH risk factors, including a survey of self-reported WASH access and use, history of deworming and direct observation of WASH facilities and participants' use of footwear. Participants in the baseline prevalence assessment provided a single stool sample for analysis. Consenting participants were provided clean stool collection containers and asked to provide a sample. For individuals unable to provide a sample at the time of container distribution, participants were instructed to collect the next day's first stool and study staff returned the following morning to collect stool samples.

Stool samples were delivered to local study laboratories in each site on the day they were collected. Each sample was analyzed by Kato-Katz[12, 13], with results for each helminth species (*Ascaris lumbricoides*, *Trichuris trichiura* and hookworm) reported in eggs per gram (epg). Two slides were read from each sample and a subset of 10% of slides was randomly selected for quality assurance by the laboratory managers. STH infection was defined as the presence of eggs on either or both of the slides read by laboratory technicians, and intensity of positive samples was calculated as an average of available reads. All samples were tracked from receipt to read using barcodes and data were entered electronically using SurveyCTO on Android phones.

## Statistical analysis

Initial data management was conducted and descriptive statistics generated using Stata 14.0 (Stata Corp, College Station, Texas). The estimation of key epidemiological parameters was performed using R 3.6.1 [14].

Study site, household and individual characteristics were summarized using numbers and percentages or medians and interquartile ranges (IQR), as appropriate. Baseline prevalence and intensity of infection were weighted to the age distribution and different cluster sizes in the baseline census, to account for the stratified sampling approach. Cluster-specific age-weighted prevalences were calculated based on age distribution in each cluster, excluding infants (who were not eligible to be sampled and therefore had no prevalence data) and those with missing ages. The final prevalence estimates were weighted for cluster size. Error bars represent weighted Agresti-Coull 95% confidence intervals.

Moderate-intensity infections are defined as 5,000–49,999 eggs per gram (epg) of stool for *A. lumbricoides*, 1,000–9,999 epg for *T. trichiura* and 2,000–3,999 epg for Hookworms. Heavy-intensity infections are defined as 50,000+ epg for *A. lumbricoides*, 10,000+ epg for *T. trichiura* and 4,000+ epg for hookworms.

We estimated key epidemiological parameters for hookworm infection. These parameters include the force of infection (FOI) and the degree of parasite aggregation within the human population (as measured by the negative binomial parameter *k*). The FOI, defined as the per host rate of worm acquisition per unit of time, was calculated for each DeWorm3 study site. The FOI results from exposure to infective stages in the environment and is a strong indicator of who is most at risk of acquiring new worms and also of the underlying mechanisms of transmission that are at work in the population. We constructed a model of worm burden as a function of age based on acquisition of worms due to infection (at a rate defined by the FOI) and the loss of mature worms due to death. We estimated FOI and its credible interval as a function of age, assuming that worm burdens at DeWorm3 study sites at baseline were approximately at equilibrium. The degree of parasite aggregation within the human host population as measured inversely by the negative binomial parameter k, was estimated using prevalence (P) and mean egg intensity (E) as follows:

$$P(E; k, \lambda) = 1 - \left( \frac{k}{k + E/\lambda} \right)^k$$

Here, $\lambda$ is the expected (mean) detected egg count per worm and $k$ is the aggregation parameter of the negative binomial distribution (a low k value indicates a highly aggregated worm distribution). Details of the likelihood, model assumptions and method of estimation can be found in S1 Text.

Most morbidity due to STH occurs in individuals with heavy intensity infections[2]. To estimate the burden of morbidity within the DeWorm3 study sites at baseline, we derived the relationship of the mean egg count from individual egg count data (for both hookworm and *A. lumbricoides*), assuming a negative binomial distribution of the egg counts (the aggregation parameter was also fitted). We then combined the fitted relationship with demographic data from the baseline census of each of the sites to estimate the total egg count output produced by the population, stratified into 5 year age categories. The total egg count in an age category serves as a proxy for the total worm burden (the mean worm burden of each individual times the number of individuals in the age category). As such, it offers insight into what fraction of the total worm burden is subject to treatment in an age-targeted round of MDA. Additionally, we estimated the probability of heavy infection in an individual within each age category, as a proxy for morbidity, using the WHO definitions for heavy infections for each STH species [15].

### Ethical considerations

Written informed consent (or oral consent with documented thumbprint in the presense of a witness) was obtained from each head of household or other adult for participation in the census and each census update. Participants in the baseline data and stool collection activities who were of age to provide consent provided written informed consent (or oral consent with documented thumbprint in the presense of a witness). If a child under the age of consent was selected for any of the above activities, parents or guardians provided informed consent on behalf of the child and children provided assent in accordance with national ethical guidelines.

The DeWorm3 Project was reviewed and approved by the Institut de Recherche Clinique au Benin (IRCB) through the National Ethics Committee for Health Research, Ministry of Health in Benin, The London School of Hygiene and Tropical Medicine, The College of Medicine Research Ethics Committee in Malawi and the Christian Medical College Institutional Review Board in Vellore, India and the Human Subjects Division at the University of Washington. The trial was registered at ClinicalTrials.gov (NCT03014167).

## Results

### Study sites

Each of the three study sites, in Benin, Malawi and India, is characterized by distinct demographic, social, cultural and epidemiologic features. The sites range in geographic size from 148 km$^2$ (Benin) to 477 km$^2$ (India) and in population from 94,969 to 140,932 (Table 1). Population density also varied across the sites. In both Malawi and India, a large proportion of the population lived in less populated rural areas, where the population density was <250 persons/km$^2$ (45.0% and 54.8% respectively). This is in contrast to Benin, where a large proportion (37.9%) of individuals lived in dense urban environments (population density ≥1000+ persons/km$^2$ (Table 1). The populations in all three sites were generally of poor socio-economic status, although there were significant differences in asset ownership and access to resources both within and between sites. For example, a very high proportion of dwellings in Malawi had unimproved flooring (79.6%) compared to Benin and India (17.6% and 12.5% respectively). Access to improved water and sanitation also differed substantially by site. Open defecation was reported by the majority of respondents in India (65.1%) and by a large proportion

**Table 1. Results of the DeWorm3 baseline census.**

| | India | | Malawi | | Benin | |
|---|---|---|---|---|---|---|
| **Study site characteristics** | **N (%)** | | **N (%)** | | **N (%)** | |
| Geographic area of study site (km$^2$) | 477 | | 289 | | 148 | |
| Total number of households | 36,536 | | 27,750 | | 24,378 | |
| Population density | | | | | | |
| <50 persons/km$^2$ | 1,710 | (4.7) | 809 | (2.9) | 840 | (3.4) |
| 50–249 persons/km$^2$ | 18,288 | (50.1) | 11,691 | (42.1) | 3,983 | (16.3) |
| 250–999 persons/km$^2$ | 14,289 | (39.1) | 15,243 | (54.9) | 10,277 | (42.2) |
| ≥1000+ persons/km$^2$ | 2,206 | (6.0) | 0 | (0.0) | 9,249 | (37.9) |
| Missing | 43 | (0.1) | 7 | (0.0) | 29 | (0.1) |
| **Household characteristics** | N = 36,536 | | N = 27,750 | | N = 24,378 | |
| | n (%) / median (IQR) | | n (%) / median (IQR) | | n (%) / median (IQR) | |
| Household size | 4 | (3–5) | 4 | (3–6) | 4 | (2–5) |
| Owner-occupied dwelling | 36,632 | (89.3) | 23,959 | (86.4) | 15,086 | (61.9) |
| Wall materials | | | | | | |
| -Natural | 7,264 | (19.9) | 1,154 | (4.2) | 5,797 | (23.8) |
| -Manmade | 29,253 | (80.1) | 26,545 | (95.7) | 18,184 | (74.6) |
| -Other / don't know / refused | 19 | (0.1) | 51 | (0.2) | 397 | (1.6) |
| Roofing materials | | | | | | |
| -Natural | 4,036 | (11.1) | 16,846 | (60.7) | 1,423 | (5.8) |
| -Manmade | 32,484 | (88.9) | 10,889 | (39.2) | 22,882 | (93.9) |
| -Other / don't know / refused | 16 | (0.0) | 15 | (0.1) | 73 | (0.3) |
| Flooring materials | | | | | | |
| -Natural | 4,564 | (12.5) | 22,077 | (79.6) | 4,300 | (17.6) |
| -Manmade | 31,932 | (87.4) | 5,656 | (20.4) | 19,976 | (81.9) |
| -Other / don't know / refused | 40 | (0.1) | 17 | (0.1) | 102 | (0.4) |
| **WaSH access** | | | | | | |
| -Sanitation | | | | | | |
| -Basic facilities | 11,674 | (32.0) | 18,882 | (68.0) | 5,673 | (23.3) |
| -Limited facilities | 653 | (1.8) | 6,970 | (25.1) | 6,727 | (27.6) |
| -Unimproved facilities | 440 | (1.2) | 1,193 | (4.3) | 2,442 | (10.0) |
| -No facilities (open defecation) | 23,769 | (65.1) | 705 | (2.5) | 9,536 | (39.1) |
| -Drinking water source | | | | | | |
| -Basic | 33,882 | (92.7) | 20,795 | (74.9) | 20,164 | (82.7) |
| -Limited | 1,143 | (3.1) | 6,483 | (23.4) | 1,553 | (6.4) |
| -Unimproved | 1,250 | (3.4) | 411 | (1.5) | 2,553 | (10.5) |
| -Surface water | 261 | (0.7) | 61 | (0.2) | 108 | (0.4) |
| **Study population** | N = 140,932 | | N = 121,819 | | N = 94,969 | |
| Female | 70,620 | (50.1) | 64,333 | (52.8) | 49,080 | (51.7) |
| Age distribution | | | | | | |
| -Infants (<1 years) | 1,750 | (1.2) | 4,368 | (3.6) | 2,616 | (2.8) |
| -Preschool-age children (1–4 years) | 8,482 | (6.0) | 17,455 | (14.3) | 11,188 | (11.8) |
| -School age children (5–14 years) | 21,839 | (15.5) | 37,652 | (30.9) | 26,043 | (27.4) |
| -Adults (15+ years) | 108,861 | (77.2) | 62,161 | (51.0) | 54,882 | (57.8) |
| -Missing | 0 | (0.0) | 183 | (0.2) | 240 | (0.3) |
| Education (school-aged children) | | | | | | |
| -Attending school | 20,594 | (94.3) | 33,772 | (89.7) | 20,031 | (76.9) |
| -Not attending school | 1,242 | (5.7) | 3,842 | (10.2) | 3,510 | (13.5) |

(*Continued*)

**Table 1.** (Continued)

| Study site characteristics | India | | Malawi | | Benin | |
|---|---|---|---|---|---|---|
| | **N (%)** | | **N (%)** | | **N (%)** | |
| -Missing | 3 | (0.0) | 38 | (0.1) | 2,502 | (9.6) |
| Highest level of education (adults) | | | | | | |
| -No education | 32,807 | (32.8) | 21,692 | (34.9) | 17,272 | (31.7) |
| -Incomplete primary school | 6,283 | (6.3) | 24,894 | (40.2) | 8,067 | (14.8) |
| -Complete primary school | 8,184 | (8.2) | 1,619 | (2.6) | 1,172 | (2.1) |
| -Some secondary school | 34,442 | (34.4) | 4,324 | (7.0) | 7,377 | (13.5) |
| -Greater than secondary school | 17,835 | (17.8) | 0 | (0.0) | 5,646 | (10.3) |
| -Missing / don't know / refused | 500 | (0.5) | 9,178 | (14.8) | 15,028 | (27.5) |
| Migration | | | | | | |
| -Lived outside the household the majority of the past 6 months | 3,788 | (2.7) | 4,630 | (3.8) | 1,416 | (1.5) |
| -Slept elsewhere the night before the census | 7,779 | (5.5) | 6,741 | (5.5) | 2,550 | (2.7) |

of respondents in Benin (39.1%), while open defecation in Malawi was low (2.5%). In contrast, India had the highest proportion of households with access to basic drinking water sources (92.7%, as compared to 82.7% in Benin and 74.9% in Malawi), and Benin had the highest proportion of unimproved or surface water sources (10.5% compared to 1.5% in Malawi and 3.4% in India). School attendance among SAC was also highest in India (94.3%) and lowest in Benin (76.9%). Similarly, a higher proportion of adults in India had completed more than a primary school education than in the other two sites (Table 1).

## Study population

There were substantial differences in the population age distribution by site, with the highest proportion of infants and young children in Malawi and the highest proportion of adults in India (Fig 2). There were 8,600 individuals in Malawi (7.1% of the population) and 1,151 individuals in Benin (1.2% of the population) who were not able to provide a precise date of birth or age and are not included in the age pyramids. The numbers of males identified in the DeWorm3 census in Malawi between the ages of 15 and 39 was less than expected based on published national demographic age profiles, possibly representing migration of young males to other areas for work or educational opportunities[16, 17].

## Prevalence and intensity of infection

The measured prevalence of any STH infection was highest in India (17.0%), followed by Malawi (7.4%) and Benin (5.3%). Across all three of the DeWorm3 study sites, hookworm was the predominant STH detected. *Ascaris* and *Trichuris* infections were rare in India and Malawi, although *Ascaris* made up approximately 40% of STH infections in Benin. (Table 2). After accounting for the stratified sampling approach by age and cluster, the weighted baseline prevalence of hookworm in India was 21.4% (95% CI: 20.4–22.4%), while prevalence of *Ascaris* and *Trichuris* by Kato-Katz was low (0.1% and 0.3% overall, respectively), The weighted prevalence of moderate- or heavy-intensity infection was 1.5% (95% CI: 1.2–1.8%) and all but one infection was caused by hookworm (1.5%). In Benin, the overall weighted prevalence was significantly lower (5.6%, 95% CI: 5.1–6.2%) and *Ascaris* (2.0%, 95% CI: 1.6–2.3%) was more common than in other sites. *Ascaris* infections were more likely to be moderate- or heavy-intensity (43.7%, unweighted) compared to hookworm (5.0%). *Trichuris* infections were rare (0.1%). In Malawi, the weighted overall STH prevalence was 7.7% (95% CI: 67.1–8.4%) and

**Table 2. Unweighted prevalence and intensity of infection data by site and species.**

| | Any STH: n (%) | Hookworm: n (%) | *Ascaris*: n (%) | *Trichuris*: n (%) |
|---|---|---|---|---|
| **India (N = 6,089)** | | | | |
| *Kato-Katz prevalence*[1] | 1,033 (17.0) | 1,012 (16.6) | 6 (0.1) | 17 (0.3) |
| *Intensity of infection*[2], *among positive Kato-Katz tests* | | | | |
| Light-intensity | 960 (92.9) | 940 (92.9) | 6 (100.0) | 16 (94.1) |
| Moderate-intensity | 50 (4.8) | 49 (4.8) | 0 (0.0) | 1 (5.9) |
| Heavy-intensity | 23 (2.2) | 23 (2.3) | 0 (0.0) | 0 (0.0) |
| **Malawi (N = 6,136)** | | | | |
| *Kato-Katz prevalence*[1] | 453 (7.4) | 436 (7.1) | 3 (<0.1) | 15 (0.2) |
| *Intensity of infection*[2], *among positive Kato-Katz tests* | | | | |
| Light-intensity | 435 (96.0) | 418 (95.9) | 3 (100.0) | 15 (100.0) |
| Moderate-intensity | 6 (1.3) | 6 (1.4) | 0 (0.0) | 0 (0.0) |
| Heavy-intensity | 12 (2.6) | 12 (2.8) | 0 (0.0) | 0 (0.0) |
| **Benin (N = 6,139)** | | | | |
| *Kato-Katz prevalence*[1] | 324 (5.3) | 199 (3.2) | 126 (2.1) | 5 (0.1) |
| *Intensity of infection*[2], *among positive Kato-Katz tests* | | | | |
| Light-intensity | 258 (79.6) | 189 (95.0) | 71 (56.3) | 4 (80.0) |
| Moderate-intensity | 54 (16.7) | 4 (2.0) | 50 (39.7) | 0 (0.0) |
| Heavy-intensity | 12 (3.7) | 6 (3.0) | 5 (4.0) | 1 (20.0) |

[1] Positivity was defined as the presence of eggs on one of two slides read by laboratory technicians. In Benin 6,136/6,139 had two slides read, in India 6,088/6,089 Kato-Katz had two slides read, and in Malawi 6,117/6,136 Kato-Katz tests had two slides read.

[2] Light-intensity infections are defined as 1–4,999 eggs per gram (epg) of faeces for *Ascaris* infection, 1–999 epg for *Trichuris* and 1–1,999 epg for Hookworms. Moderate-intensity infections are defined as 5,000–49,999 epg for *Ascaris*, 1,000–9,999 epg for *Trichuris* and 2,000–3,999 epg for Hookworms. Heavy-intensity infections are defined as 50,000+ epg for *Ascaris*, 10,000+ epg for *Trichuris* and 4,000+ epg for Hookworms.

was mostly driven by hookworm (7.4%). *Ascaris* (0.1%) and *Trichuris* (0.3%) infections were rare in Malawi and there were very few moderate- or heavy-intensity infections (0.3%, 95% CI: 0.2–0.4%) (Fig 3).

Given the very low rates of *A. lumbricoides* and *T. trichiura* infection in India and Malawi, these were excluded from this analysis. However, given that a substantial proportion of STH infections in Benin were due to *A. lumbricoides*, we did include infections in the analysis. Projections of the modelled negative binomial distribution for egg counts suggest that the total hookworm egg counts by age, and, by proxy, the total worm burden by age, were highest in adult age classes in India, where counts peaked between 20 and 50 years of age. In India, the total egg count ranged up to a maximum approximate mean of 17,000 in the 45–50 year old age-category (Fig 4A). In Malawi, the highest total egg counts were found in adolescents (maximum approximate mean of 5,000 in the 15–20 year age-category, Fig 4C) suggesting the burden in Malawi was approximately a third of that observed in India. The total cluster egg counts in Benin were the lowest of the three countries (maximum approximate mean of 2,500), indicating worm burdens approximately 15% of the maximum found in India. The total egg counts in Benin were relatively constant up to 40 years of age, dropping gradually to almost 0 in the oldest age category (Fig 4E). In all three sites, the probability of heavy hookworm infection increased with increasing age (Fig 4B, 4E and 4F). The observed relationships between the mean cluster EPG and cluster prevalence suggest a negative binomial distribution of parasite numbers per host (as reflected in egg output) with differing degrees of aggregation depending on age and country (Fig 5B, 5D and 5F). The highest *k* value (i.e. the lowest degree of aggregation) was observed in India, which showed higher values of infection prevalence and mean

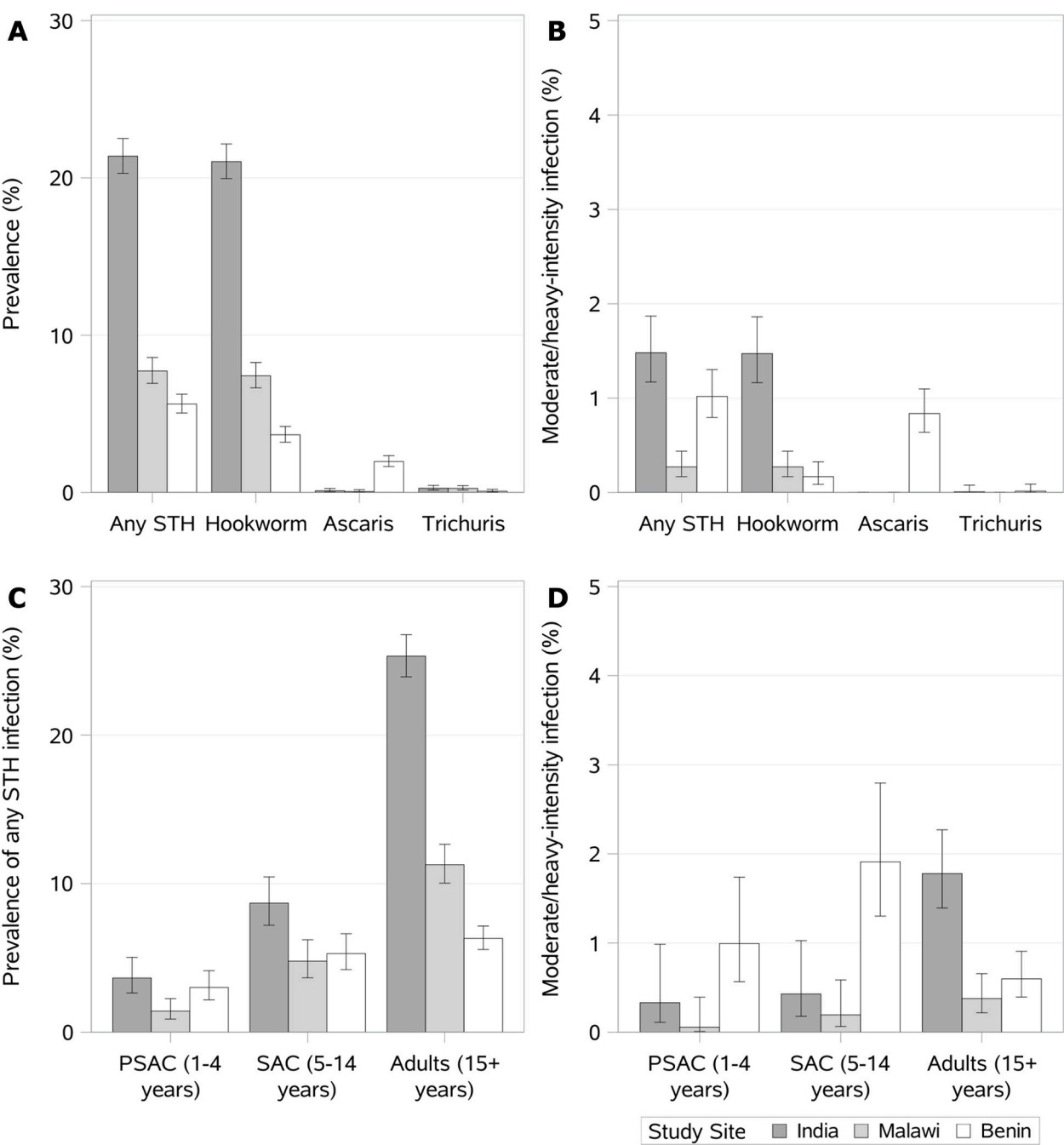

**Fig 3.** Age- and cluster-weighted (A) prevalence estimates of infection and (B) prevalence of moderate- or heavy-intensity of infection. Weighted estimates account for the stratified sampling approach by age (stage 1) and cluster (stage 2). Cluster-specific age-weighted prevalences ($Pr_{cw}$) were calculated by taking sum of the age-specific prevalences ($Pr_{PSAC}$, $Pr_{SAC}$ or $Pr_{adult}$) multiplied by the proportion (Prop) of that age group for each cluster among those 1+ years old ($Pr_{cw} = Pr_{PSAC}{}^*Prop_{PSAC}+Pr_{SAC}{}^*Prop_{SAC}+Pr_{Adult}{}^*Prop_{Adult}$). The final age- and cluster-weighted prevalences ($Pr_w$) were calculated by taking the sum of the cluster-specific age-weighted prevalences multiplied by the proportion of the census population living in each cluster ($Pr_w = Pr_{c1w}{}^*Prop_{c1}+ Pr_{c2w}{}^*Prop_{c2} . . . Pr_{c40w}{}^*Prop_{c40}$). Error bars represent weighted Wilson 95% confidence intervals with design effect adjustment.

cluster EPG (Fig 5B) compared with the other two countries. Malawi and Benin showed high levels of aggregation (reflected by small *k* values) and low levels of infection prevalence (Fig 5D

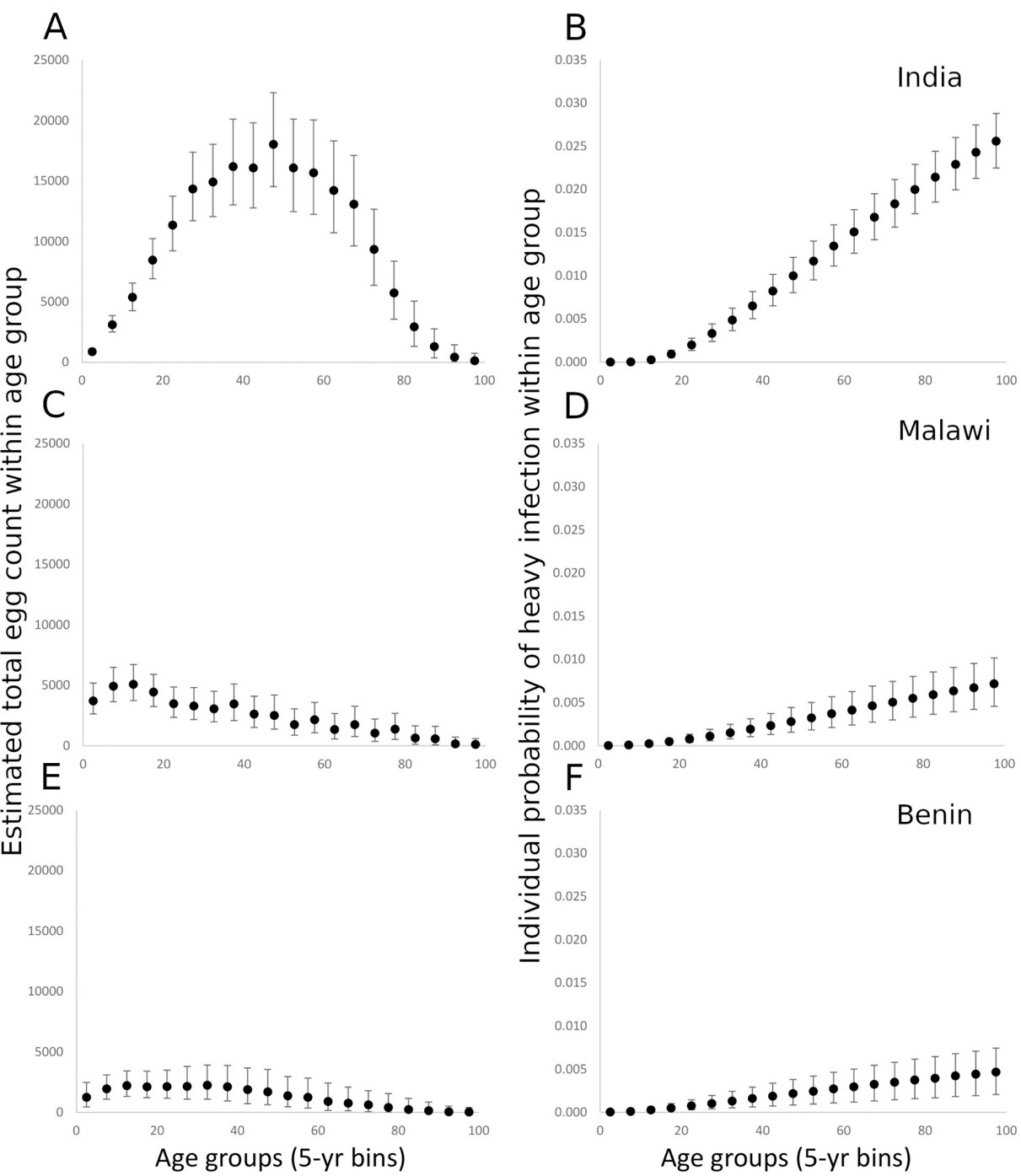

**Fig 4.** Fig 4A, C and E record the estimated total hookworm egg count in the human host population by 5-year age categories, projected from the negative binomial probability distribution for egg counts fitted to the sample data for India (A), Malawi (C) and Benin (E). Fig 4B, 4D and 4F record the estimated individual-level probability of heavy hookworm infection by 5-year age categories, calculated from the negative binomial distribution for egg counts fitted to the sample data for India (B), Malawi (D) and Benin (F). All error bars represent 95% credible intervals.

and 5F). The $k$ values are estimated at 0.17 (95% CI 0.14–0.21), 0.027 (95% CI 0.020–0.035) and 0.025 (95% CI 0.017–0.037) for India, Malawi and Benin respectively (Table 3).

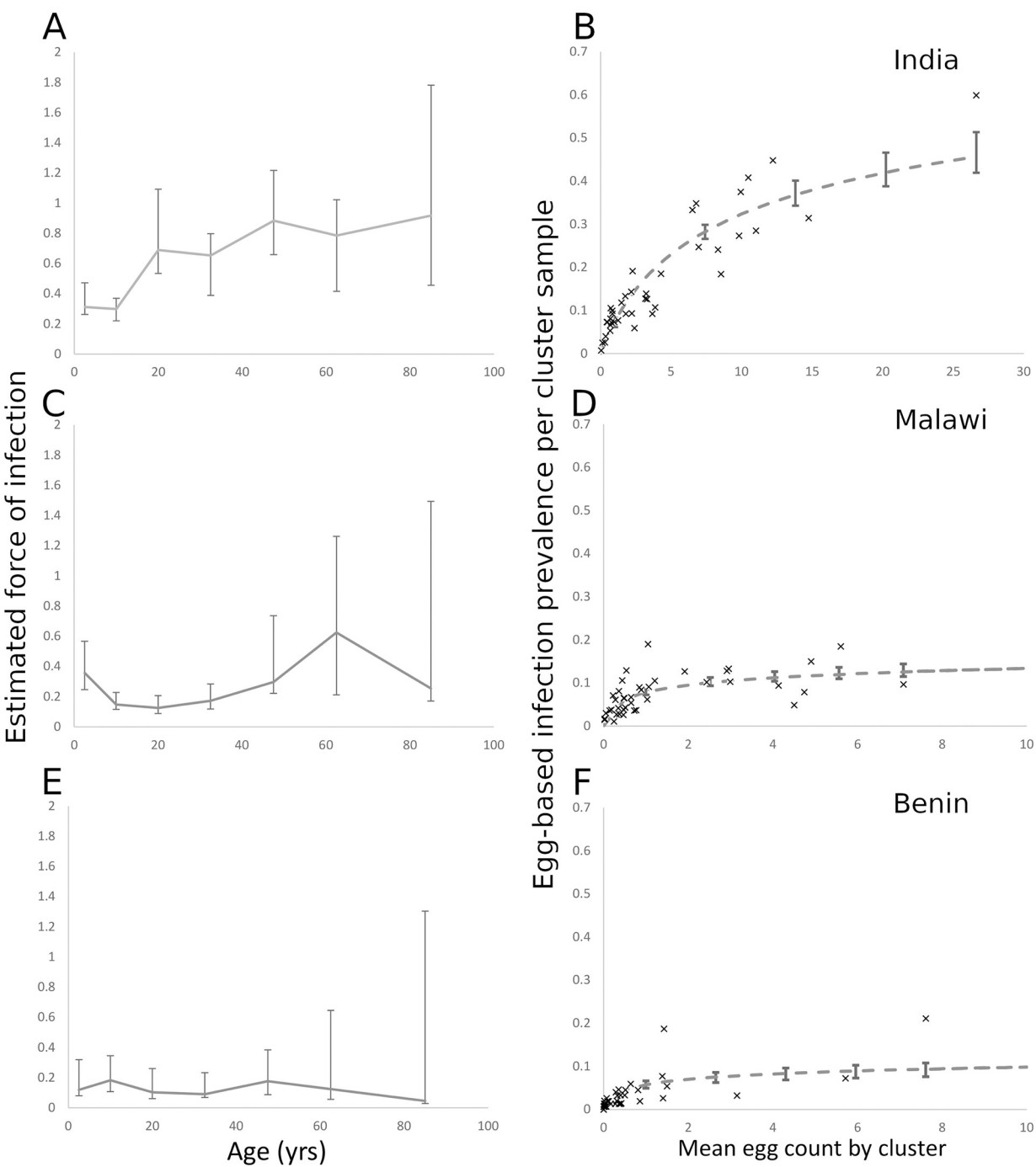

**Fig 5.** Fig 5A, 5C and 5E record the force of infection for hookworm (the per host rate of acquisition of worms per unit of time (year)) as a function of age, calculated from individual hookworm egg count data for India (A), Malawi (C) and Benin (E). Fig 5B, 5D and 5F show the cluster prevalence of infection plotted against the hookworm mean egg count for individual cluster samples. The curve represents the best fit of a negative binomial probability distribution relating the prevalence of infection to the mean egg count (S1 Text).

The FOI for hookworm was highest in adults in India and Malawi but appeared relatively stable across age groups in Benin (Fig 5A, 5C and 5E). There was a significant dip in the relative FOI among school-age children in India (5–14 years), which may reflect the impact of past

**Table 3. Aggregation parameter estimates (*k*) and their 95% credible intervals and mean egg count per worm coefficient, $\lambda$.**

|  | Aggregation parameter (*k*) (95% credible interval) | Mean egg count per worm ($\lambda$) (95% credible interval) |
|---|---|---|
| India | 0.17 (0.14, 0.21) | 4.8 (3.8, 6.1) |
| Malawi | 0.027 (0.02, 0.035) | 1.18 (0.53, 1.9) |
| Benin | 0.025 (0.017, 0.037) | 2.73 (1.36, 4.1) |

rounds of MDA that have been delivered through the schools (Fig 5A). The Malawi infection patterns show a marked decrease in the FOI across a much broader age range (10–40 years of age). Benin is characterized by a very flat FOI profile across age groups. There is no significant difference in the per capita rate of infection between age groups (Fig 5E). Unfortunately, because the prevalence was so low for *A. lumbricoides* in Benin, it was not possible to precisely calculate the total egg counts by age and morbidity for this parasite. The estimated FOI of *A. lumbricoides* in Benin that was calculated showed the highest FOI among adults (40–60 years, Fig 6A). As with hookworm, there was a high degree of aggregation of *A. lumbricoides* infection in the human population (*k* = 0.14 95% CI 0,11–0.17, Fig 6B).

## Discussion

The DeWorm3 project is a large, multicountry community cluster randomized trial assessing the feasibility of interrupting the transmission of STH globally. The size of this study (over 357,000 individuals enrolled in the baseline census), the geographic and epidemiologic diversity across the study sites, the harmonized data collection efforts and the robust analytical methods utilized in DeWorm3 will provide rigorous data to inform the development of future treatment strategies and guidelines for STH. Results presented here highlight distinct differences in the profiles of the three DeWorm3 sites, including differences in demographic characteristics, socioeconomic status and environment, including altitude, average temperatures, vegetation cover and rainfall. There are also significant cultural and societal differences within and between the three country sites. This diversity in site conditions and populations is an important characteristic of the DeWorm3 trial, as it facilitates a nuanced understanding of the many external barriers and facilitators to STH transmission interruption.

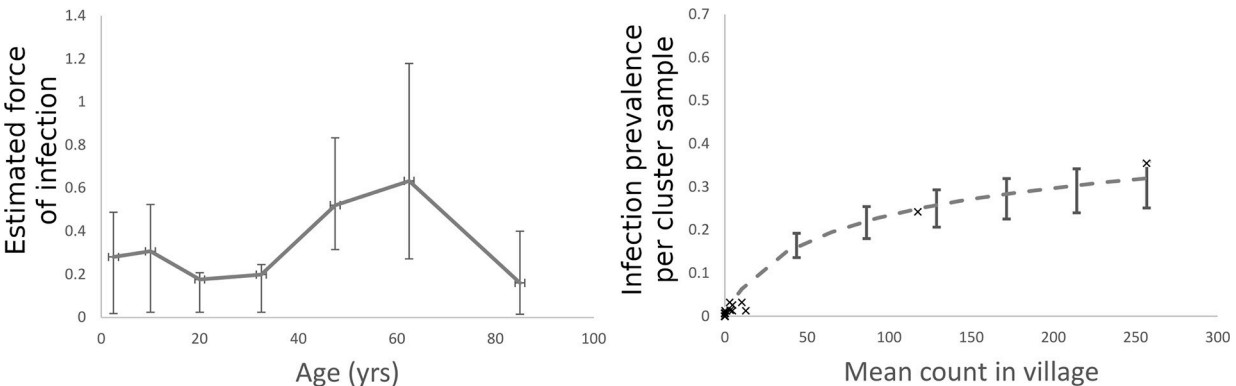

**Fig 6.** Fig 6A records the force of infection for Ascaris (the per host rate of acquisition of worms per unit of time (year)) as a function of age, calculated from individual Ascaris egg count data for Benin. Fig 6B shows the cluster prevalence plotted against the Ascaris mean egg count for individual cluster samples. The curve represents the best fit of a negative binomial probability distribution relating the prevalence of infection and the mean egg count (S1 Text).

Baseline prevalence of STH varied substantially across DeWorm3 sites. While prevalence in India was 21.4%, prevalence in Benin and Malawi was <10%, and species-specific prevalence in Benin, the only site with a substantial *Ascaris* prevalence, was even lower at ~2%. Sensitivity of Kato Katz is low at low prevalence and intensity of infection, meaning that these prevalences are likely to be underestimated[18]. In addition, all of the study sites completed at least five rounds of mass drug administration of albendazole as treatment for lymphatic filariasis prior to study initiation. As such, this previous intensive treatment would be expected to transiently reduce prevalence/intensity of STH in these populations. It is also important to note that both Ascaris and Trichuris are most highly prevalent in younger children and adolescents while hookworm prevalence peaks in adulthood. As a result, targeted treatment of school-age and preschool age children would be expected to disproportionately impact infection in these age groups and therefore in these species[19].

The epidemiology of STH infection and transmission is driven by many environmental, behavioural and demographic factors[20–22]. The demographic characteristics in Benin and Malawi demonstrate high birth rates, where young children are the largest group in the population. India exhibits a demographic profile with a smaller proportion of young children and a greater proportion of young adults. This demography is especially important across the sites in DeWorm3 where hookworm is the dominant infection, as older age groups typically have higher average intensities of infection. This is supported by the data from the India DeWorm3 site, where we saw a higher prevalence and intensity of STH infection, driven predominantly by hookworm in adults.

The age and sex distributions observed in all of the DeWorm3 study sites also reveal reduced numbers of young male individuals as compared to estimates from national census statistics[16]. This is especially marked in Malawi and may reflect migration of young males out of the study communities in search of work elsewhere. Migration is a common feature of many low-income countries and may impact the ability of stand-alone MDA programs to interrupt STH transmission. If individuals migrate to areas with comparable or higher prevalence of infection, and return periodically to the study area, they may be absent during MDA treatment and/or re-introduce infection into the community when they return[23].

The baseline DeWorm3 parasitological data indicate that the distribution of worms at the individual level appears highly aggregated, where most people harbor few worms and a few people harbor many[24]. The majority of observed hookworm infections in these populations are of light intensity, although there was a strong predisposition to heavy infection observed for *A. lumbricoides* in Benin, likely due to a combination of socio-economic, behavioural and genetic risk factors[25, 26]. A number of recent epidemiological studies have recorded increased parasite aggregation as prevalence falls in value under MDA impact, which may reflect patterns of compliance to treatment[27]. Access to clean water and improved sanitation may also drive some of these differences in infection status. For example, India has the highest rates of open defecation and the highest baseline prevalence of infection. Future analyses will evaluate the key factors influencing individual infection status and evidence for predisposition to heavy infection at each site.

The total egg output in a given section of the population serves as a proxy for the total worm burden in that group and highlights the importance of adults in driving the continued transmission of these infections in many settings. The distribution is calculated based on the individual mean intensities of infection in an age group and the size of that age group in a given population. In India, the age profile for the total worm burden is characterized by a pattern where most infection is harboured by adults, with most of the worms found in individuals in the age range 30–70 years of age. A similar shape is seen in the Benin data, but with the peak between 10–40 years of age. This is a consequence of the youngest age groups having lower mean intensities of infection, whilst the older age groups, which generally have high intensity

infections, represent a smaller proportion of the overall population. In contrast, the Malawi data show a monotonic decline with age with the highest worm burdens in the individuals less than 20 years of age. This reflects the younger population at the Malawi site, with an especially large proportion falling into the youngest age bands (Fig 2). The patterns observed in all three sites reflects the typical pattern of hookworm infection observed in most regions of the world with endemic infection, where a rate of infection almost independent of age results in the highest burdens accumulating in the adult age groups[28, 29].

The probability of heavy infection in each age group is approximately a linear function of age in all countries, with the oldest individuals having the highest probability of heavy infection. However, the oldest age groups represent a relatively small proportion of the overall population at each site. In this analysis, the rate of increase in the probability of heavy infection with age is approximately 5 times greater in India than in Malawi and Benin, reflecting the higher prevalence and intensity of infection found in the India study site. There is also a significant dip in the relative burden of infection among SAC in India, likely reflective of past and ongoing biannual school-based deworming. In the Malawi data, there is a strong drop in FOI between the ages of approximately 10 and 40, which may also be the result of the impact of long-term age-targeted deworming among school-age children spreading to older age groups with time, or a result of past treatment of all age groups through the community-wide lymphatic filariasis MDA program.

National preventive chemotherapy programs currently implemented within the study sites differ greatly in terms of targeted population, frequency, format (school-based vs deworming days) and past treatment coverage. All three countries in the DeWorm3 trial delivered MDA for both STH and lymphatic filariasis control prior to the collection of these baseline data. As such, the patterns of infection documented do not reflect the pristine transmission intensity in each region. Historical treatment coverage matters greatly and where data are available, back calculation methods can be employed to ascertain the pristine transmission intensity (ie. the transmission intensity in an endemic setting prior to introduction of control methods)[30]. However, in the absence of historical treatment coverage, the baseline prevalence and intensity of infection in each age group provides valuable information regarding necessary program inputs for interrupting STH transmission. The pristine FOI in any given location (as measured by the basic reproductive number, $R_0$, which is a direct quantitative measure of transmission intensity) determines how quickly individuals and communities reacquire infection following a round of MDA, and as such determines the frequency and coverage of MDA required to interrupt transmission[31–33]. The results of the baseline DeWorm3 data are in line with estimates from prior models suggesting that interruption of transmission may be achievable within clusters at all three sites with community-wide MDA provided at high coverage and compliance[6].

There are several important limitations to the epidemiologic profiling presented. First, this paper reports Kato Katz faecal egg counts as measures of the intensity of infection in individuals, which has several implications. As mentioned previously, Kato Katz is known to be unreliable at low prevalences of infection, tending to underestimate the true prevalence[18]. This is likely to amplify the observed levels of over-dispersion, as sensitivity is greater for moderate and heavy intensity infections. The use of egg counts to estimate the worm aggregation parameter value is a further limitation. The values estimated are higher than those typically seen in the literature[27]. This may be because the use of egg counts compounds (i) the aggregation of worms within individuals with (ii) the aggregated nature of the egg output in stool, leading to higher estimated aggregation relative to direct measurement from worm expulsion studies [34]. As a result, the level of aggregation as measured from egg counts is likely to be higher than if it had been measured directly via worm expulsion studies.

The large credible intervals observed around the relative FOI profiles are expected given the large variance to mean ratios arising from the highly aggregated negative binomial distributions of egg intensity measures. Indeed, the variability seen in the mean values across the age profile are generally of a similar size as the 95% credible intervals, suggesting a lack of a strong age-related profile of infection. Real-time polymerase chain reaction (PCR) will be performed on stored stool samples from all DeWorm3 sites to to detect STH DNA and will enable ascertainment of this potential bias. Future publications will report study results based on this more sensitive diagnostic tool. An additional limitation was that a high proportion of the population in the Benin and Malawi study sites were unable to report an exact age or date of birth. Finally, the census data obtained in DeWorm3 relied heavily on self-report, which may have led to reporting bias.

The DeWorm3 trial is designed to assess the feasibility of interrupting the transmission of STH across a range of epidemiologic transmission settings. The study has demonstrated significant social, demographic, environmental and geographic differences between the three countries participating in this multi-country community cluster randomized trial. These differences provide important baseline data used to determine the age specific intensity of transmission (the FOI) at each site. In addition, each site provides important comparative data to both assess the impact of the MDA in the trial sites and to further our understanding of STH epidemiology in settings where elimination of STH may be feasible. At the conclusion of this large multi-country randomized trial, these data will enable global policy makers to assess the feasibility of interrupting the transmission of STH and to determine the specific contextual factors likely to facilitate success in the elimination of these infections in specific geographic areas.

## Supporting information

**S1 Text. Supplementary information.**
(DOCX)

**S2 Text. The DeWorm3 Trials Team Author List.**
(DOCX)

## Acknowledgments

We acknowledge the dedication and effort of the all members of the communities participating in the DeWorm3 study as well as the DeWorm3 teams in Benin, India and Malawi. We also acknowledge the members of the DeWorm3 Data Safety and Monitoring Committee; Richard Hayes, Harriet Mpairwe, Purushothaman Jambulingam, Jimmy Whitworth and the DeWorm3 Scientific Advisory Group; Dean Jamison, Zulfiqar Bhutta and Nilanthi de Silva. We would also like to acknowledge Antonio Montressor, Wilfred Batcho, Achille Massougbodji, Dorothée Gazard, Manfred Accrombessi, Rajeshkumar Rajendiran, Chinnaduraipandi Paulsamy, Sobana Devavaram, Samuel Paul Gideon Martin, Noel Joyce Mary Hillari, Naveen Kumar Sekar, Dhanalakshmi Manoharan, Selvi Laxmanan, Dasthagir Basha, Alvin Blessings Chisambi, Zac Kamwendo, Alfred Mbwinja, Lyson Samikwa, Leanne Doran, Victoria Carter, Steven Williams, Nils Pilotte, Katherine Thomas, Barbra Richardson, Alyson Shumays, Simon Brooker, Kelli Jehle and Christy Hanson.

## Author Contributions

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
