## [Decision Letter · Decision Letter 0]

29 Apr 2020

Dear Dr. Walson,

Thank you very much for submitting your manuscript "Baseline patterns of infection in regions of Benin, Malawi and India seeking to interrupt transmission of soil transmitted helminths (STH) in the DeWorm3 trial" for consideration at PLOS Neglected Tropical Diseases. As with all papers reviewed by the journal, your manuscript was reviewed by members of the editorial board and by several independent reviewers. In light of the reviews (below this email), we would like to invite the resubmission of a significantly-revised version that takes into account the reviewers' comments. 

We cannot make any decision about publication until we have seen the revised manuscript and your response to the reviewers' comments. Your revised manuscript is also likely to be sent to reviewers for further evaluation.

Sincerely,

Adam Akullian, Ph.D.

Associate Editor

Mar Siles-Lucas

Deputy Editor

Please ensure review 1's comments (attached below) are included in the response.

Comments from Editor:

The authors report "age-adjusted" prevalence for each site in the abstract and results, but only mention methods for calculating "age-weighted" prevalence. Please ensure consistency in naming convention for prevalence estimates. If estimates were further adjusted by age to compare across sites please specify in the methods. If not please remove "Age-adjusted" terminology as it would indicate that estimates are comparable across sites with respect to similar age-distributions (which it seems is not the case here). Instead, it appears estimates are weighted by age to ensure they reflect the age-distribution within each site.

Reviewer's Responses to Questions

**Key Review Criteria Required for Acceptance?**

**Methods**

-Are the objectives of the study clearly articulated with a clear testable hypothesis stated?

-Is the study design appropriate to address the stated objectives?

-Is the population clearly described and appropriate for the hypothesis being tested?

-Is the sample size sufficient to ensure adequate power to address the hypothesis being tested?

-Were correct statistical analysis used to support conclusions?

-Are there concerns about ethical or regulatory requirements being met?

Reviewer #1: (No Response)

Reviewer #2: This study presents baseline infection data and epidemiological parameters, including host statistics, from a large scale study on three main soil transmitted helminths. The study is informative, in that it covers large areas from three countries, but recapitulates what generally known on the patterns of infection of these three helminths in low/medium income countries. 

This is a very data rich study and it would be useful to look at these data in more detail and address some of the speculations reported in the Discussion. For example, comments are made on a possible effect of gender or sanitation level on the intensity of infection, however, it seems to me that these data are available (table 1) and analyses can be straightforward. Similarly, it will be useful to examine if there is any spatial pattern in the dynamics of infection and host data, and their relationships, within each country. Are there any high risk areas? And, Are these areas the same for all the three helminths? Given the large variation in some of those trends, I am wondering if this is a consequence of combining the data together. 

The work needs some attention on formatting, typos, legend description, and particularly, clarify of the sections. In this respect, Methods need to be explained and developed, while it is good to refer to previous studies, a clarity in what is available, from where and when, what is used and how is used in the different analyses will help to move through smoothly. There is some repetition in the text and some sentences seems to contradict each other. As previous noted, more can be done with the data and Results should be definitely improved. Finally, it will be useful to put this work in a broader context and elaborate more on it in the introduction and discussion.

**Results**

-Does the analysis presented match the analysis plan?

-Are the results clearly and completely presented?

-Are the figures (Tables, Images) of sufficient quality for clarity?

Reviewer #1: (No Response)

Reviewer #2: Please, see above

**Conclusions**

-Are the conclusions supported by the data presented?

-Are the limitations of analysis clearly described?

-Do the authors discuss how these data can be helpful to advance our understanding of the topic under study?

-Is public health relevance addressed?

Reviewer #1: (No Response)

Reviewer #2: Please, see above

**Editorial and Data Presentation Modifications?**

Reviewer #1: (No Response)

Reviewer #2: Please, see above

**Summary and General Comments**

Reviewer #1: (No Response)

Reviewer #2: Please, see above

PLOS authors have the option to publish the peer review history of their article (what does this mean?). If published, this will include your full peer review and any attached files.

Reviewer #1: Yes: Assaf P. Oron

Reviewer #2: No
---

## [Decision Letter · Decision Letter 1]

3 Sep 2020

Dear Dr. Walson,

We are pleased to inform you that your manuscript 'Baseline patterns of infection in regions of Benin, Malawi and India seeking to interrupt transmission of soil transmitted helminths (STH) in the DeWorm3 trial' has been provisionally accepted for publication in PLOS Neglected Tropical Diseases.

Best regards,

Adam Akullian, Ph.D.

Associate Editor

Mar Siles-Lucas

Deputy Editor

Reviewer's Responses to Questions

**Key Review Criteria Required for Acceptance?**

**Methods**

-Are the objectives of the study clearly articulated with a clear testable hypothesis stated?

-Is the study design appropriate to address the stated objectives?

-Is the population clearly described and appropriate for the hypothesis being tested?

-Is the sample size sufficient to ensure adequate power to address the hypothesis being tested?

-Were correct statistical analysis used to support conclusions?

-Are there concerns about ethical or regulatory requirements being met?

Reviewer #1: Yes yes, yes..... yes and lastly, no ethical concerns.

The confidence intervals still seem generally too narrow to me, but for a baseline survey CIs are arguably far less important than the point estimates, so I'm fine with things remaining as they are.

**Results**

-Does the analysis presented match the analysis plan?

-Are the results clearly and completely presented?

-Are the figures (Tables, Images) of sufficient quality for clarity?

Reviewer #1: Yes

**Conclusions**

-Are the conclusions supported by the data presented?

-Are the limitations of analysis clearly described?

-Do the authors discuss how these data can be helpful to advance our understanding of the topic under study?

-Is public health relevance addressed?

Reviewer #1: Yes

**Editorial and Data Presentation Modifications?**

Reviewer #1: No

**Summary and General Comments**

Reviewer #1: Thank you for your thoughtful addressing of all comments. I can't wait to see the study results, and hope that the MDAs have not been disrupted substantially by the pandemic.

Good luck!

PLOS authors have the option to publish the peer review history of their article (what does this mean?). If published, this will include your full peer review and any attached files.

Reviewer #1: **Yes: **Assaf Peretz Oron

---

## [Editor Report · Acceptance letter]

16 Oct 2020

Dear Dr. Walson,

We are delighted to inform you that your manuscript, "Baseline patterns of infection in regions of Benin, Malawi and India seeking to interrupt transmission of soil transmitted helminths (STH) in the DeWorm3 trial," has been formally accepted for publication in PLOS Neglected Tropical Diseases.

Best regards,

Shaden Kamhawi

co-Editor-in-Chief

Paul Brindley

co-Editor-in-Chief
